# KCS-FCnet: Kernel Cross-Spectral Functional Connectivity Network for EEG-Based Motor Imagery Classification

**DOI:** 10.3390/diagnostics13061122

**Published:** 2023-03-16

**Authors:** Daniel Guillermo García-Murillo, Andrés Marino Álvarez-Meza, Cesar German Castellanos-Dominguez

**Affiliations:** Signal Processing and Recognition Group, Universidad Nacional de Colombia, Manizales 170003, Colombia

**Keywords:** functional connectivity, kernel methods, motor imagery, EEG, cross-spectral distribution, deep learning

## Abstract

This paper uses EEG data to introduce an approach for classifying right and left-hand classes in Motor Imagery (MI) tasks. The Kernel Cross-Spectral Functional Connectivity Network (KCS-FCnet) method addresses these limitations by providing richer spatial-temporal-spectral feature maps, a simpler architecture, and a more interpretable approach for EEG-driven MI discrimination. In particular, KCS-FCnet uses a single 1D-convolutional-based neural network to extract temporal-frequency features from raw EEG data and a cross-spectral Gaussian kernel connectivity layer to model channel functional relationships. As a result, the functional connectivity feature map reduces the number of parameters, improving interpretability by extracting meaningful patterns related to MI tasks. These patterns can be adapted to the subject’s unique characteristics. The validation results prove that introducing KCS-FCnet shallow architecture is a promising approach for EEG-based MI classification with the potential for real-world use in brain–computer interface systems.

## 1. Introduction

The Media and Information Literacy (MIL) methodology proposed by UNESCO encompasses a range of essential competencies crucial for individuals to actively participate in human development, as highlighted in recent studies [1]. Among the most notable technological advancements in this field is the emergence of Brain-Computer Interfaces (BCIs), which have the potential to revolutionize human-technology interaction, particularly for individuals with motor disabilities [2]. Motor Imagery (MI) is a widely-used BCI paradigm that refers to the mental rehearsal of motor tasks without any physical movement [3]. Figure 1 depicts a conventional experimental setup for MI, including brain activity signal acquisition, commonly through Electroencephalography (EEG), stimuli delivery and marker synchronization using a visual cue, and posterior data processing. The use of MI is beneficial in a wide range of contexts, such as enhancing language comprehension in children and older adults [4], neurofeedback training therapy for individuals with Parkinson’s disease [5], and improving attentional focus [6]. Moreover, specific brain activities elicited by cognitive processes, i.e., MI tasks, must be identified. Then, techniques such as Event-Related Potentials (ERPs) will contain particular patterns for different processes [7]. In other words, ERP gather time-frequency information about different brain areas [8]. However, it is essential to note that MI’s Signal-to-Noise Ratio (SNR) can be significantly affected by other background brain processes and artifacts [9].

EEG is a non-invasive and portable neuroimaging technique that measures brain electrical activity over the scalp at a high temporal resolution, reflecting synchronized oscillatory activity. As a result, EEG is commonly used to measure brain activity evoked by MI [11]. Yet, during data acquisition, the limited spatial information provided by EEG data and its susceptibility to various disturbances, such as electromyographic signals from the heart, eyes, tongue, and muscles, can lead to a low SNR and decrease the accuracy of MI-BCI [12]. In addition, classifying EEG records is challenging due to their high dimensionality and non-stationary behavior. Moreover, ERP components recorded on the scalp via EEG reflect the sum of superimposed components generated from different neural sources. Hence, the observed patterns reflect distinct neural sources, misleading the estimation of time-frequency features when cognitive processes or stimuli are elicited by brain activity [8,13]. Therefore, developing robust and accurate signal processing methods is crucial to improve time-frequency feature estimation in MI.

There are several strategies for enhancing BCI performance and developing MI skills. One practical approach is subject-specific models, as each individual may have unique self-regulation evoked responses in diverse frequency bands [14]. As a result, most MI-BCI systems are based on subject-specific temporal and spectral features [15,16], typically calculated on a single-trial basis. One example is the Filter Bank-Common Spatial Patterns (FBCSP) method, which leverages task-related brain rhythms primarily localized in the sensorimotor area [17]. Besides, brain functional connectivity is often used to measure synchronization within sensorimotor rhythms [18]. Furthermore, many approaches with high performance and interpretability rely on covariance matrices computed from the EEG signal. Nevertheless, due to the high nonlinearity, intra-class variability inherent in MI tasks, and the small training data size, standard Euclidean distance-based covariance estimation tends to be less accurate [19].

MI-BCI systems are generally enhanced using different frequency features from an array of overlapped bandpass filters. Still, the ad hoc selection of the bandwidth and overlap percentage are some of the main drawbacks, resulting in low temporal resolution, overfitting, and complex tuning. Nonetheless, advancements in the field of Deep Learning (DL) have helped to tackle these issues. In this light, Convolutional Neural Networks (CNN)-based techniques have surpassed classical machine learning and signal processing methods in extracting relevant local and general spatio-spectral information from raw EEG [20]. Indeed, temporal and spatial CNN blocks are widely used in DL to extract relevant feature maps at the top of the model [21,22,23].

However, applying DL-based models to achieve reliable single-trial MI classification remains challenging. First, relevant spatial-temporal-spectral feature maps are desired to overcome overfitting due to high intra-class variability [24,25]. Second, more robustness in dealing with noise in the raw EEG while avoiding complex architecture is required [26,27]. Third, DL lacks straightforward interpretability, which is critical to validate neural activity for medical diagnosis, monitoring, and computer-aided learning [28,29,30]. Therefore, additional efforts are needed to improve DL-based approaches for EEG-based MI-BCI, aiming for richer spatial-temporal-spectral feature maps, yielding more straightforward and explainable algorithms.

This paper presents a groundbreaking technique for classifying MI using EEG signals, termed Kernel Cross-Spectral Functional Connectivity Network (KCS-FCnet). Our approach overcomes current DL limitations by introducing a cross-spectral Gaussian functional connectivity data-driven estimator to classify MI tasks from raw data. KCS-FCnet utilizes 1D convolutional layers to extract temporal-frequency features from input channels and a cross-spectral distribution estimation that codes relevant temporal and spectral MI patterns. It also includes a functional connectivity feature map, which improves the interpretability of the model by extracting meaningful pairwise channel relationships. Our approach is evaluated on a publicly available dataset and achieves state-of-the-art results for EEG-based MI classification. Furthermore, it demonstrated robustness to different experimental settings and individual differences. Lastly, our results suggest that the KCS-FCnet architecture is a highly effective method for EEG-based MI classification and can potentially be applied in real-world BCI.

The remainder is organized as follows: Section 2 and Section 3 describe the related work reviewed along with the materials and methods. Section 4 and Section 5 present the experiments and discuss the results. Finally, Section 6 outlines the conclusions and future work.

## 2. Related Work

The gold standard for MI-based BCI classification is the FBCSP method, which relies on multiple bandpass filters, varying overlap percentages, and bandwidth to extract features [17]. This strategy aims to extract more relevant information by leveraging subject-dependent brain rhythms. However, CSP-based approaches are sensitive to noise, and overfitting [31]. Thus, several regularized CSP-derived approaches have been proposed to overcome these issues, which can be split into cost function and covariance regularization. In addition, the L2-norm is used in optimization problems (e.g., Tikhonov regularization CSP and weighted Tikhonov regularized CSP), but outlier data and artifacts degrade its performance. To cope with this issue, L1-norms (e.g., CSPL1 and SFBCSP) are employed to find sparse spatial features that are more representative [32]. Other strategies like shrinkage are also used to estimate covariance matrices, especially under small sample sizes accurately; the simplest way to reduce the variance of the covariance estimator is to apply diagonal loading. A more sophisticated strategy is proposed in [33] where a shrinkage estimator asymptotically minimizes the mean square error, being well-conditioned for small sample sizes. Moreover, it can be applied to high-dimensional problems. Although interpretable and efficient algorithms are proposed to improve the calculation of covariance matrices, their efforts are limited and tend to become less effective against severe covariance changes within the same brain state (either left or right).

Regarding DL, different strategies have been developed. For example, authors in [34] use a deep belief network with a conditional random field to recognize emotions; deep belief networks are strong feature extractors since they can learn complex nonlinear functions. Nevertheless, those networks usually require an unsupervised pre-training stage, which is sensitive to a small training set. On the other hand, CNNs encode raw EEG signals or topographic plots (TGP), showing high-performance accuracy. For instance, 1D CNNs over raw EEG can extract adaptive temporal-frequency features involving spatial information [35]. Likewise, 2D CNNs can codify spatial information over a fixed bank of TGP calculated with different time windows and frequency filters [36]. Nonetheless, 2D CNNs get local information depending on their size, so they can be affected when information is not centered on a specific spatial location. Furthermore, the temporal dependency of EEG channels has been decoded using Long-Short Term Memory (LSTM) approaches, such as authors in [37]. LSTMs can find an implicit representation of temporal sequences, but their training stage requires exhaustive fine-tuning and an expensive computational burden.

Besides, DL approaches rely on the hypothesis that architectures will learn invariant, generalizable features constrained by the number of available data [38]. For instance, Shallowconvnet proposed in [21] uses two main convolution layers. The first is designed to bandpass each EEG channel, while the second is analogous to spatial filters. Moreover, it employs a square and log activation function, an average pooling layer, and a fully connected layer to emulate the FBCSP strategy [39]. In addition, Deepconvnet, also proposed by [21], uses similar ideas as Shallowconvnet but operates three additional convolutional layers to extract deeper features. Nevertheless, complex DL models are prone to overfitting by capturing trial-specific variability rather than actual patterns in the same neural states, either left or right. In contrast, authors in [22] proposed the EEGnet, which utilizes depthwise separable convolutions to reduce the number of training parameters, lowering network complexity while yielding significant temporal and spatial features. TCFussionnet is a more recent approach proposed by [23] that comprises three main parts. Like EEGnet, it includes a temporal component that allows for learning different bandpass frequencies and a depth-wise separable convolution to extract spatial features for each temporal filter. A Temporal Convolutional Network (TCN) block is also applied to extract temporal patterns, which are then concatenated with the output of a separable convolution to alleviate feature loss. These outputs are then flattened and concatenated, followed by a separable convolution. A final dense layer with softmax activation is used to classify the concatenated features into MI classes. Similarly, adversarial learning has been successfully applied in many DL applications; these architectures rely on training a generative model that enforces invariance and generalization. Regardless, adversarial architectures are affected by overfitting, especially with subjects that do not have a similar pattern in each session [38].

Overall, DL approaches try to solve the class variability from a different point of view and attempt to generate more generalized models, reaching better MI-BCI performance results. However, most solutions still need direct interpretability compared with classical covariance-based approaches that take advantage of the calculated functional connectivity. In this paper, we leverage the end-to-end KCS-FCnet on functional connectivity to keep direct interpretability and on CNNs to extract more relevant feature maps from raw EEG.

## 3. Materials and Methods

The present study introduces a cutting-edge technique for extracting spatial-temporal-frequency information specific to the subject from EEG recordings. Individuals tend to engage in MI tasks within varying spectral ranges, so it is crucial to devise a strategy that effectively captures this relevant information while preserving the less prominent frequency bandwidths. To accomplish this, we propose using the kernel cross-spectral distribution, a powerful tool that offers valuable insights into nonlinear pairwise channel relationships. Furthermore, this innovative approach allows for identifying subject-specific discriminant features within a DL framework, which can significantly enhance the accuracy and precision of MI-BCI. Following, the tested dataset and the main mathematical background are presented in detail.

### 3.1. Motor Imagery Dataset

We propose to validate our approach using the Giga dataset, which is a comprehensive collection of MI-EEG recordings obtained from 52 subjects. However, for the purpose of our study, we will only consider 50 subjects who met the minimum requirement of having at least 100 EEG trials recorded. The Giga dataset is an ideal choice for our validation as it has been widely used in the field of MI classification and has been shown to provide a robust benchmark for evaluating the performance of different models [40].

The Giga dataset is recorded using a 10–10 placement electrode system with 64 channels, which is a widely used setup for MI-EEG recordings. The electrode system is configured to record brain activity during a trial, as illustrated in Figure 2 (first row). Each channel is recorded for 7 s and is sampled at 512 Hz, providing a high-resolution representation of the neural activity. Before each run, each subject is asked to perform a finger movement task, starting with the index finger and proceeding to the little finger, touching each finger to their thumb within 3 s after the onset of the task. This task is used as a means to calibrate the subject’s MI, ensuring that the subsequent EEG recordings accurately reflect the subject’s brain activity.

Besides, a fixation cross was displayed on a black screen at the beginning of each trial for 2 s, signaling the start of the task. Following the fixation cross, a cue was presented, linking either to the right or left hand MI label. The cue instructed the subjects to imagine touching each of their fingers of the selected hand with the thumb, starting from the index finger and proceeding to the little finger. To ensure that the subjects were imagining the kinesthetic experience of the task, and not the visual experience, they were instructed to focus on the sensation of finger movement. This task was repeated for 3 s. At the end of each trial, a break period was indicated by a blank screen that randomly lasted between 2.1 and 2.8 s. The procedure described above was repeated 20 times to complete a single run, and each subject performed between five and six runs in total. Furthermore, a cognitive questionnaire was carried out after each run. Finally, to ensure the quality of the data, each trial was tested and classified as a “bad trial” based on the voltage magnitude. Any flawed trials present in each subject were excluded from the analysis. Additionally, along with the runs set, a single-trial resting-state was recorded, lasting 60 s. The entire procedure is illustrated in Figure 2 (second row).

### 3.2. Kernel-Based Cross-Spectral Distribution Fundamentals

Let x∈X be a stochastic process that is wide-sense stationary and has a real-valued auto-correlation function $R_{x}\left(\tau \right)\in R$, as follows [41]:(1)$$R_{x}\left(\tau \right)=\int_{R}\exp\left(j2\pi \tau f\right)dP_{x}\left(f\right),$$
where $P_{x}\left(f\right)\in \left[0,1\right]$ is a monotonic, absolutely continuous, and differentiable spectral distribution for frequency f∈R.

Now, let us consider two records, $x,x^{\prime }\in R^{N_{t}}$, the univariable relationship in Equation (Equation 1) that operates over positive-definite functions can be expanded to a pairwise correlation between the random vectors x, $x^{\prime }$ through the use of a generalized, stationary kernel, $\kappa \;:\;R^{N_{t}}\;\times \;R^{N_{t}}\;\to \;R$. This kernel maps the input space into a Reproducing Kernel Hilbert Space (RKHS) through a nonlinear function ϕ:X→H. However, this expansion is only possible if the following assumption holds between both spectral representations [42]:(2)$$\kappa \left(x-x^{\prime }\right)=\int_{\Omega }\exp\left(j2\pi {\left(x-x^{\prime }\right)}^{⊤}f\right)S_{xx^{\prime }}\left(f\right)df,$$
where f∈Ω is a vector-valued frequency domain containing the bandwidth set Ω, and $S_{xx^{\prime }}\left(f\right)\in C$ is the cross-spectral density function, holding $S_{xx^{\prime }}\left(f\right)=dP_{xx^{\prime }}\left(f\right)/df$, where $P_{xx^{\prime }}\left(f\right)\in \left[0,1\right]$ is the cross-spectral distribution. As seen in Equation (Equation 2), the cross-spectral distribution within the specified bandwidth Ω can be computed using the stationary kernel κ(·), yielding:(3)$$P_{xx^{\prime }}\left(\Omega \right)=2\int_{\Omega }F\left\{\kappa \left(x-x^{\prime }\right)\right\}df.$$
where F{·} stands for the Fourier transform and $P_{xx^{\prime }}\left(\Omega \right)\in \left[0,1\right]$ gathers the cross-frequency information between x and $x^{\prime }$ within the bandwidth set Ω by extracting nonlinear data dependencies using κ(·). This approach allows us to capture the nonlinear interactions between different EEG channels, providing a more accurate representation of the underlying neural activity. Additionally, the use of a stationary kernel ensures that the proposed method is able to capture the temporal dynamics of EEG signals, which is crucial for the accurate classification of MI.

In this sense, the Gaussian kernel is a popular choice in pattern analysis and machine learning due to its ability to approximate any function and its mathematically tractable properties [43]. This makes it an ideal alternative for computing the Kernel-based Cross-Spectral distribution fixing the following Gaussian function:(4)$$\kappa_{G}\left(x-x^{\prime };\sigma \right)=\exp\left(\frac{-{∥x-x^{\prime }∥}_{2}^{2}}{2\sigma^{2}}\right),$$
where ${∥\cdot ∥}_{2}$ is the squared Euclidean distance and $\sigma \in R^{+}$ is a scale parameter. The use of a Gaussian function in Equation (Equation 3) allows for effective and efficient computation of nonlinear interactions between x and $x^{\prime }$.

### 3.3. Kernel Cross-Spectral Functional Connectivity Network

The input-output EEG dataset, ${\left\{X_{r}\in R^{N_{c}\times N_{t}},y_{r}\in {\left\{0,1\right\}}^{N_{y}}\right\}}_{r=1}^{R}$, comprises *R* trials, $N_{t}$ time instants, $N_{c}$ channels, and $N_{y}$ classes. To enhance the most informative EEG spatial-temporal-spectral patterns from $X_{r}$ and reduce noise for improved MI class prediction, we propose to estimate the cross-spectral distribution among channels using a function composition. This approach gathers 1-D convolutional-based feature layers for extracting time-frequency patterns within each EEG channel, and a Gaussian kernel-based pairwise similarity, as follows:(5)$${\hat{P}}_{r}\left(w_{f}\right)=\kappa_{G}\left(\cdot ;\sigma \right)\circ \varphi \left(X_{r};w_{f}\right),$$
where notation ∘ stands for function composition, $\varphi (\cdot ;w_{f})$ is a 1-D convolutional layer that can be used to automatically extract frequency patterns ruled by the weight vector $w_{f}\in R^{\Delta_{t}}$, with $\Delta_{t}<N_{t}.$ Of note, in Equation (Equation 6) operates the $\kappa_{G}\left(\cdot \right)$ for each pair of filtered EEG channels regarding the weights $w_{f}$; then, ${\hat{P}}_{r}\;\left(w_{f}\right)\;\in \;{\left[0,1\right]}^{N_{c}\times N_{c}}$. Moreover, we can stack a set of frequency filters to compute an average functional connectivity measure among EEG channels, as follows:(6)$${\tilde{P}}_{r}\left(\hat{\Omega }\right)=E_{f}\;\left\{\kappa_{G}\left(\cdot ;\sigma \right)\circ \varphi \left(X_{r};w_{f}\right);\forall f\in \left\{1,2,\ldots ,N_{f}\right\}\right\}$$
where $N_{f}$ is the number of 1-D convolutional filters and $\hat{\Omega }$ is the boosted frequency bandwidth concerning ${\tilde{P}}_{r}\left(\hat{\Omega }\right)\in {\left[0,1\right]}^{N_{c}\times N_{c}}$. This measure provides a way to analyze how different frequency bands of a single-trial EEG relate to each other across channels.

After computing the average functional connectivity measure, a straightforward softmax-based output layer is applied over a vectorized version of ${\tilde{P}}_{r}\left(\hat{\Omega }\right)$, which is the upper triangular matrix. This takes advantage of the symmetric property of the Gaussian functional connectivity, meaning that ${\tilde{P}}_{r}^{cc^{\prime }}\left(\hat{\Omega }\right)={\tilde{P}}_{r}^{c^{\prime }c}\left(\hat{\Omega }\right)$, $\forall c,c^{\prime }\in \left\{1,2,\ldots ,N_{c}\right\}$. Then, MI class membership can be predicted as:(7)$${\hat{y}}_{r}=\mathrm{softmax}(v⊗\mathrm{vec}({\tilde{P}}_{r}\left(\hat{\Omega }\right))+b),$$
where $v\in R^{(N_{c}\left(N_{c}-1\right)/2\times N_{y}}$, $b\in R^{N_{y}},$, ${\hat{y}}_{r}\in {\left[0,1\right]}^{N_{y}},$ and ⊗ stands for tensor product. In addition, a gradient descent-based framework using back-propagation is employed to optimize the parameter set $\Theta =\{w_{f},v,b,\sigma ;\forall f\in \left\{1,2,\cdots ,N_{f}\right\}\},$ as follows [44]:(8)$$\Theta^{*}=\underset{\Theta }{\arg\;\min}\;E_{r}\;\left\{L\left(y_{r},{\hat{y}}_{r}|\Theta \right);\forall r\in \left\{1,2,\ldots ,R\right\}\right\},$$
being L{·} a given loss function, i.e., cross-entropy. The optimization problem outlined in Equation (Equation 8) enables the training of our Kernel Cross-Spectral Functional Connectivity Network (KCS-FCnet) for the classification of MI tasks.

## 4. Experimental Set-Up

### 4.1. KCS-FCnet Implementation Details

In this study, we evaluate the efficacy of our proposed method for extracting subject-specific functional connectivity matrices from the KCS-FCnet that predicts MI output labels from EEG records. To accomplish this, we have developed a pipeline consisting of the following steps, which were tested on the Giga dataset (as detailed in Section 3.1):–Raw EEG Preprocessing: First, we load subject recordings using a custom databases loader module (https://github.com/UN-GCPDS/python-gcpds.databases (accessed on 27 January 2023)). Next, we downsample each signal from 512 Hz to 128 Hz using the Fourier method provided by the SciPy signal resample function (https://docs.scipy.org/doc/scipy/reference/generated/scipy.signal.resample.html (accessed on 27 January 2023)). Then each time series trial was filtered between [4, 40] Hz, using a fifth-order Butterworth bandpass filter. In addition, we clipped the records from 0.5 s to 2.5 s post cue onset, retaining only information from the motor imagery task. Preprocessing step resembles the one provided by authors in [22]. Note that since we are analyzing only the MI time segment, we assume the signal to be stationary. Our straightforward preprocessing aims to investigate five distinct brain rhythms within the 4 to 40 Hz range, including theta, alpha, and three beta waves. Theta waves (4–8 Hz), located in the hippocampus and various cortical structures, are believed to indicate an “online state” and are associated with sensorimotor and mnemonic functions, as stated by authors in [45]. In contrast, sensory stimulation and movements suppress alpha-band activity (8–13 Hz). It is modulated by attention, working memory, and mental tasks, potentially serving as a marker for higher motor control functions. Besides, tested preprocessing also comprises three types of beta waves: Low beta waves (12–15 Hz) or “beta one” waves, mainly associated with focused and introverted concentration. Second, mid-range beta waves (15–20 Hz), or “beta two” waves, are linked to increased energy, anxiety, and performance. Third, high beta waves (18–40 Hz), or “beta three” waves, are associated with significant stress, anxiety, paranoia, high energy, and high arousal.–KCS-FCnet Training: We split trials within each subject data using the standard 5-fold 80–20 scheme. That means shuffling the data and taking 80% of it to train (training set), holding out the remaining 20% to validate trained models (testing set), and repeating the process five times. For the sake of comparison, we calculate the accuracy, Cohen’s kappa, and the area under the ROC curve to compare performance between models [46,47]. It is worth noting that we rescale the kernel length according to the new sampling frequency as in [22]. The GridSearchCV class from SKlearn is used to find the best hyperparameter combination of our KCS-FCnet. The number of filters $N_{f}$ is searched within the set {2,3,4}.–Group-Level Analysis: We build a scoring matrix that contains as many rows as subjects in the dataset, 50 for Giga, and six columns, including accuracy, Cohen’s kappa, and the area under the ROC curve scores, along with their respective standard deviation. To keep the intuition of the higher, the better, and constrain all columns to be between [0,1] in the score matrix, we replace the standard deviation with its complement and normalize the Cohen’s kappa by adding to it the unit and dividing by two. Then, using the score matrix and the k-means clustering algorithm [47], with *k* set as three, we trained a model to cluster subjects’ results based on the baseline model EEGnet [22] in one of three groups: best, intermediate, and worst performing subjects. Next, we order each subject based on a projected vector obtained from the first component of the well-known Principal Component Analysis (PCA) algorithm applied to the score matrix. Next, with the trained *k*-means, the subjects analyzed by our KCS-FCnet were clustered using the score matrix. The aim is to compare and check how subjects change between EEGnet and KCS-FCnet-based groups.

A KCS-FCnet sketch can be visualized in Figure 3. The detailed KCS-FCnet architecture is summarized in Table 1. All experiments were carried out in Python 3.8, with the Tensorflow 2.4.1 API, on Google Colaboratory and Kaggle environments. The fine-tuning process for the model’s parameters begins by utilizing the training set for optimization. To evaluate the model’s performance, the test set is employed solely for reporting scores. The categorical cross-entropy loss function is applied, and no additional callbacks are utilized. The training phase involves passing the entire batch of samples. Additionally, to support further analysis and experimentation, the model weights and performance scores are systematically saved for future reference.

### 4.2. Functional Connectivity Pruning and Visualization

To compare functional connectivity between the groups mentioned above, first, we have to check which connections are relevant for class separability. It is worth noting that a high correlation in the functional connectivity matrix does not guarantee a higher class separability. Therefore, we use the two-sample Kolmogorov–Smirnov (2KS) test to overcome this issue and select only relevant connections as in [48]. The null hypothesis is that both samples are drawn from the same unknown distribution. Thus, we group the trials of each connection for a subject according to the label to build the samples “right” and “left”. Then, every pair is passed through the 2KS test, and connections holding a *p*-value equal to or lower than 0.05 are kept. Hence, we can state that both samples came from different distributions and the classes are distinguishable. Next, we build a *p*-value matrix containing the information on whether a connection is relevant. To visualize how each *p*-value matrix changes across subjects and groups, we plot each *p*-value matrix on a 2D visualization, where both dimensions are calculated using the well-known *t*-distributed Stochastic Neighbor Embedding (*t*-SNE) algorithm [49], from the SKlearn library, over the EEGnet score matrix. It is noteworthy that the perplexity parameter has been specifically set to a value of ten, while all other parameters have been retained at their default settings.

Next, to effectively depict the connections between various regions of the brain, we employ a specialized connectivity visualizer (https://github.com/UN-GCPDS/python-gcpds.visualizations (accessed on 27 January 2023)) which utilizes the Circos plot technique to display only the most significant connections, specifically those that fall within the 99-th percentile. To further enhance the analysis, we have chosen to plot the subject closest to the centroid of each group, thereby allowing for a detailed examination of one individual from each group.

### 4.3. Method Comparison

We compare the proposed KCS-FCnet with four end-to-end DL models that have been reported recently for effectively extracting relevantly explainable information from raw EEG. As with our proposal, the contrasted architectures are selected because they benefit from convolutional layers to extract temporal-frequency features for improving MI classification performance. Namely, (i) the EEGnet architecture in [22] operates depthwise separable convolutions to reduce the number of training parameters, extracting temporal and spatial convolution features from each channel of a previous feature map; (ii) Shallowconvnet in [21] incorporates two convolution layers (for sequential bandpass and spatial filtering of the EEG channels) followed by a square and log activation function, an average pooling layer, and a fully connected layer to emulate the baseline strategy of Filter Bank Common Spatial Patterns [39]; (iii) Deepconvnet proposed by [21] employs three convolutional layers to extract DL features; and (iv) TCFussionnet comprises three filtering stages to extract temporal, bandpass spectral, and spatial features, as explained in detail in [23].

For concrete testing, individual subject accuracy and standard deviation scores are only compared between the EEGnet and our KCS-FCnet due to their similarity in architecture and the number of parameters. For all provided approaches, the average classification performance along the 50 subjects in Giga is computed. Every architecture is implemented using TensorFlow2 and the SciKeras library, which allows wrapping any deep learning model as a SKlearn classifier. For the EEGnet, Shallowconvnet, Deepconvnet, and TCFussionnet, we use the hyperparameters that each work reported as the best combination. The complete codes for training, validating, and saving the model are publicly available (EEGnet (https://www.kaggle.com/dggarciam94/eegnet-11-11-2022-version (accessed on 27 January 2023)), Shallowconvnet (https://www.kaggle.com/dggarciam94/shallownet-11-11-2022-version (accessed on 27 January 2023)), Deepconvnet (https://www.kaggle.com/dggarciam94/deepconvnet-11-11-2022-version (accessed on 27 January 2023)), TCFussionnet (https://www.kaggle.com/dggarciam94/tcnet-fusion-11-11-2022-version (accessed on 27 January 2023)), and KCS-FCnet (https://www.kaggle.com/code/dggarciam94/gfcnet-11-11-2022-version (accessed on 27 January 2023)).

## 5. Results and Discussion

### 5.1. Subject Dependent and Group Analysis Results

The proposed KCS-FCnet architecture is closely compared to the EEGnet architecture in this study, with a focus on subject-specific accuracy scores and their standard deviation. The comparison is illustrated in Figure 4, where the dotted orange line represents the EEGnet and the dotted blue line represents the proposed KCS-FCnet. The blue and red bars indicate whether a specific subject’s accuracy improves or decreases when using the KCS-FCnet, respectively. Additionally, the background of the figure includes low-opacity green, yellow, and red bars to indicate the group belongingness of the subjects (best, intermediate, and worst-performing clusters). The *X*-axis of the figure displays the subjects sorted based on their maximum score values as determined by the EEGnet results. The average accuracy for EEGnet and KCS-FCnet is 69.0 and 76.4, respectively, resulting in an incremental of 7.4 for our proposal. Overall, it is demonstrated that KCS-FCnet can effectively classify motor imagery tasks using raw EEG as input data.

Moreover, our proposed method, KCS-FCnet, demonstrates mixed results in terms of accuracy for the subjects studied. On the one hand, seven subjects experienced a decrease in accuracy, of which only four experienced a reduction of three points or more. On the other hand, the remaining subjects experience an increase in accuracy, with the majority experiencing an increase of more than five points. Notably, our approach has a particularly strong impact on subjects in the third group, resulting in only one case where KCS-FCnet fails to surpass the baseline performance and two cases with less than one point of increase. Additionally, our data-driven functional connectivity method proves effective in extracting relevant feature maps for subjects in the second group, with over ten subjects experiencing an accuracy increment of at least three points. The first group, consisting of subjects with good performance, does not see remarkable results from KCS-FCnet, with only one subject experiencing a decrease in performance by five points and one subject experiencing an increase of more than five points. In general, subjects with the best performance appear to have a limitation when trying to include more relevant feature maps, yet our network is able to preserve their classification performance in most cases. In contrast, poor-performance subjects have more room for enhancement in the feature map, which is why we see a more significant increment in the third group.

Figure 5 illustrates the subject group belongingness and the impact of the KCS-FCnet method on group classification. The first row shows the subjects organized based on the EEGnet results, while the bottom row shows how each subject changes or maintains their group based on the KCS-FCnet results. For example, in the red group on the EEGnet row, subjects starting from S52, when we look at the new grouping based on KCS-FCnet for the same subset of subjects, it is evident that a total of eleven subjects significantly improved their performance, moving to the yellow cluster, while only nine remained in the red one. Additionally, six subjects had a major performance increase and were promoted to the best group (green), which demonstrates the effectiveness of the proposed framework. Furthermore, the subjects that were originally in the best group maintained their status, indicating that the best-performing subjects are less likely to improve. Then, our approach achieves better MI discrimination compared to EEGnet, particularly for bad and medium-performing subjects, which is important as it highlights the model’s capacity to handle challenging cases.

Next, Table 2 shows the accuracy results for each group for EEGnet and KCS-FCnet. It is important to note that while the difference in the first group is insignificant, with only 0.9 points, there is a notable improvement in the second group, with an increment of 5.6 accuracy points. Additionally, the third group shows a considerable increment of 12.4 points. A similar trend is observed in the standard deviation, where the second group has the most reduction of 2.6 points. Hence, our proposal not only outperforms EEGnet in terms of accuracy but also reduces the variability for all clusters.

Figure 6 compares the accuracy standard deviation for EEGnet and KCS-FCnet. The background boxes indicate the group membership. For the first group, we can see an improvement in the variability scores for our proposed method, with a difference of four points between the maximum values. The second group shows a slight reduction in all standard deviation values for our method; however, the variability proportion remains almost the same. For the last group, there is a similar behavior for both methods. Overall, our proposed strategy reduces the variability and maintains a similar average accuracy score among subjects in the best group while increasing the average accuracy and maintaining the variability for the second and third clusters.

### 5.2. Estimated Functional Connectivity Results

In this study, we employed the two-sample Kolmogorov–Smirnov test to calculate a functional connectivity matrix for each subject. The matrix includes information about the separability of the MI classes. The null hypothesis asserts that the distribution of connection pairs for classes 0 and 1 is identical. A *p*-value is calculated, and we reject the null hypothesis only if the *p*-value is less than 5%. In other words, connection pairs with lower *p*-values indicate higher class separability and are considered more informative for the classification task Figure 7 depicts the results of the test in the form of *p*-value-based matrices for each subject, which are plotted in a 2D projection using the *t*-SNE algorithm to reduce the dimensionality of the score matrix. Each matrix has a colored outer square that indicates the group membership. The matrices in the top left corner (first group) have the lowest *p*-values for every connection pair, indicating that almost every pair has a different class distribution, resulting in high accuracy scores, e.g., more than 90%. Conversely, matrices in the bottom right corner (third group) have the least significant *p*-values, indicating that only a few connection pairs can reject the null hypothesis; then, the class probability distribution for almost every pair cannot be distinguished. There is a gradual transition between matrices from the highest *p*-values in the bottom right corner to the lowest in the top left corner. Additionally, each group keeps an intra-subject *p*-value similarity for similar EEG channel connections.

Furthermore, Figure 8 details the amount of information preserved within each subject representative connectivity matrix. We utilize the widely used quadratic Rényi’s entropy [50] to quantify the interpretability performance from pruned functional connectivity matrices. Namely, a higher entropy value indicates a higher interpretability of that particular group of subjects concerning both relevant pair-wise channel relationships and MI discrimination capability. The background boxes represent the group membership, and the box-and-whisker plots depict each cluster’s distribution of Rényi’s entropy values. The first group displays the most significant values, indicating that most connections discriminate highly between classes. In contrast, the third group has the lowest values, suggesting poor class discriminability. As expected, the groups that perform better show higher retention of information by the KSC-FCnet-based functional connectivity matrix.

Further, the Circos plot is a valuable tool to visualize which EEG channels are most important for each subject’s experiment. Figure 9 shows that relevant channel dependencies are kept mainly for the best performance group. Note that all connections are normalized between the three subjects (connectivities above the 99th percentile are shown). For the G I case, the most robust connections are found between the frontal, central left, and right areas, with a few connections in the posterior region. This pattern is consistent with a good-performing subject who presents the most relevant information in the sensorimotor area (central left and right). Conversely, G II shows significant connections between the center-right and frontal areas, with fewer robust connections in the central left. It is worth noting that EEG noise may be present in the connectivity feature map around the central left region. Notably, G III has no significant connections, indicating that the model could not extract noise-free and discriminative connectivities.

The second and third rows in Figure 9 depict the most significant brain areas using 2KS and the weights of the last layer in the KCS-FCnet. In G I, similar results are observed, highlighting the sensorimotor area; however, the results in the third row are more concentrated around C4 and C3. It suggests that subjects in the best-performing group do not exhibit much noise and the MI task can be completed using only a few sensors. For G II, there is a slight difference between the results. In particular, a high activation in the left frontal area is observed, while the information is more focused on the sensorimotor area. Finally, for the last group, the most significant difference is observed. While in the 2KS test, some importance (below 0.4) is observed around C4 and C3, in the weight-based approach, there is no clear pattern, indicating that our DL approach can not find relevant information in the sensorimotor area.

### 5.3. Method Comparison Results: Average MI Classification and Network
Complexity

Table 3 summarizes the results of KCS-FCnet and the contrasted end-to-end architectures of Convolutional neural network models. Deepconvnet performs the poorest, making it unsuitable for handling high intra-class variability. By contrast, Shallowconvnet and TCFussionnet have values of quality measures that are very close to each other, being more competitive. Despite this, KCS-FCnet achieves the highest scores, outperforming the other models.

Another essential aspect to quantify the model performance is the number of trainable parameters. Figure 10 presents the required number of trainable parameters vs. the attained MI classification performance for each studied DL approach. As seen, a higher number of trainable parameters does not necessarily imply a higher classification accuracy. In fact, two clusters are evident: models holding less than 20k trainable parameters and algorithms requiring more than 100k parameters. Notably, the EEGnet gathers 2.194 trainable parameters and got a 69% accuracy score. Then, the Deepconvnet has the most significant number of trainable parameters (178.927) but only achieves a 62.5% accuracy score. The overfitting issue can explain the latter, especially when dealing with a high intra-class variability MI dataset. As previously stated, the Shallowconvnet, TCFussionnet, and KCS-FCnet have the highest accuracy scores. However, the Shallowconvnet has more than 100k trainable parameters, and the TCFussionnet has more than 25k. Conversely, our KCS-FCnet, not only outperforms these architectures in terms of accuracy but also requires the lowest complexity to achieve competitive discrimination results.

## 6. Conclusions

We have developed a cutting-edge EEG-based motor imagery classification technique known as the Kernel Cross-Spectral Functional Connectivity Network (KCS-FCnet). This method combines cross-spectral analysis to extract important features from EEG signals, and a data-driven Gaussian functional connectivity block to model the non-linear connections between different channels. We evaluated the effectiveness of our technique using a widely-used public dataset, the Giga dataset and found that it outperforms current state-of-the-art methods regarding EEG-based MI classification, spatio-temporal-frequency interpretability, and a number of trainable parameters (network complexity). Additionally, our method has the added benefit of being able to automatically estimate the cross-spectral distribution from subject-specific filters and provide interpretable functional connections. Furthermore, compared to other similar architectures, KCS-GFCnet demonstrates an impressive performance increase, and even holds its own against more complex methods, while having the second-fewest trainable parameters. Our analysis of results from three natural groups also showed that the top-performing subjects retained more critical connections than the other groups. Overall, our results indicate that the KCS-FCnet approach is a highly promising method for EEG-based MI classification and has the potential to be utilized in practical brain–computer interface applications.

In future work, the authors plan to address the issue of high intra-class variability and overfitting, specifically in the worst-performing subjects, by incorporating functional connectivity regularization based on Rényi’s entropy [51]. Additionally, further analysis of the inter-class variability will be conducted, specifically focusing on how each subject performs within different runs of the experiment. We aim to understand if subjects learn to execute motor imagery tasks as they progress through the runs and whether this results in better performance in subsequent runs [52,53]. Furthermore, we plan to analyze subject-specific filters, specifically looking at the activity of the filter waves concerning brain frequency bands such as Theta, Alpha, Beta, and Gamma. This will provide insight into how the brain processes motor imagery tasks and can further improve the performance of our method.

## Figures and Tables

**Figure 1 diagnostics-13-01122-f001:**
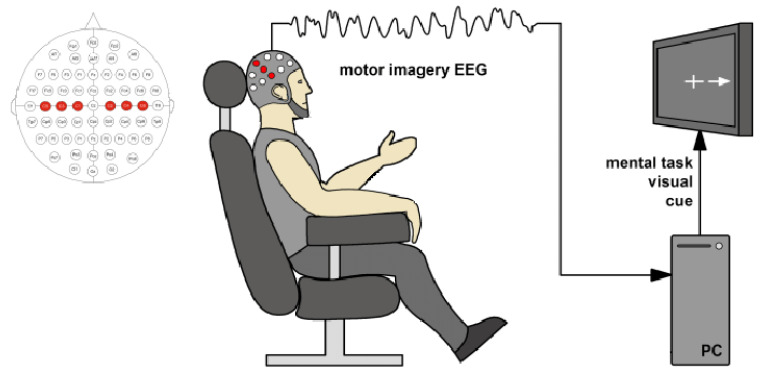
Experimental setup of validation methodology performed for Motor Imagery-based BCI. Source: Adapted from [10].

**Figure 2 diagnostics-13-01122-f002:**
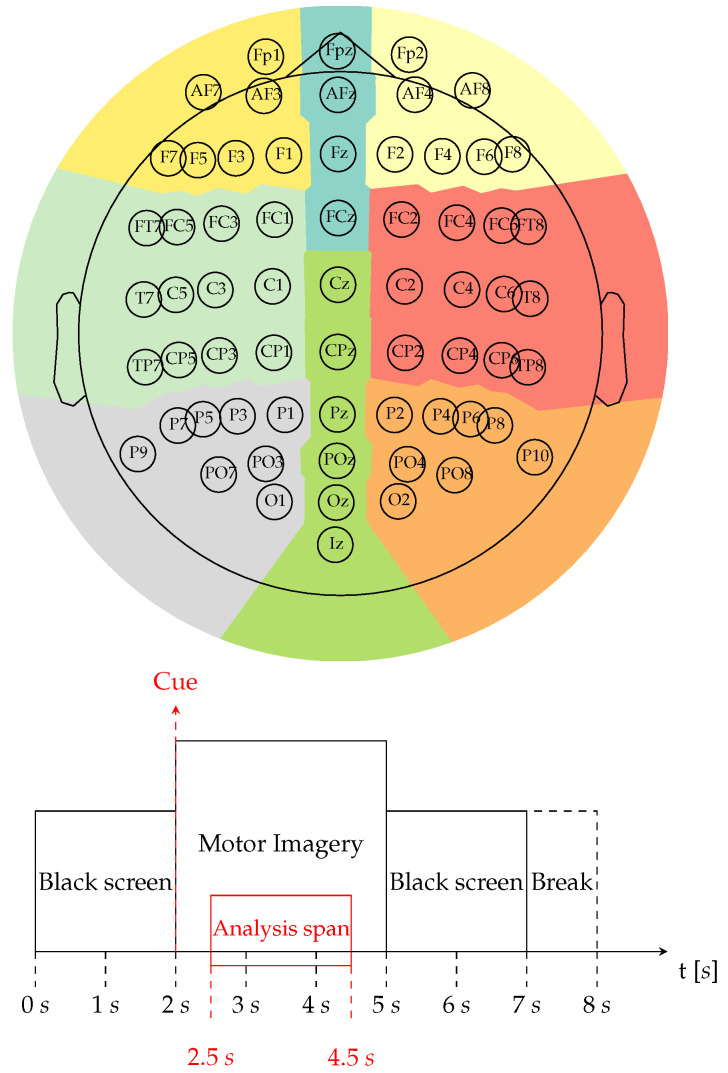
Giga dataset for Motor Imagery classification. First row: The dataset features a topoplot, which illustrates the sensor positions in a 10–10 placement electrode system, containing 64 channels. Besides, it highlights in color the main parts of the brain (
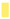
 Frontal left, 
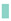
 Frontal, 
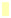
 Frontal right, 

 Central right, 
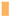
 Posterior right, 

 Posterior, 
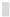
 Posterior left, 
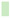
 Central left). Second row: Motor Imagery paradigm. The EEG within the interval of 2.5 to 4.5 s is used for concrete testing in the classification of Motor Imagery for left vs. right hand.

**Figure 3 diagnostics-13-01122-f003:**
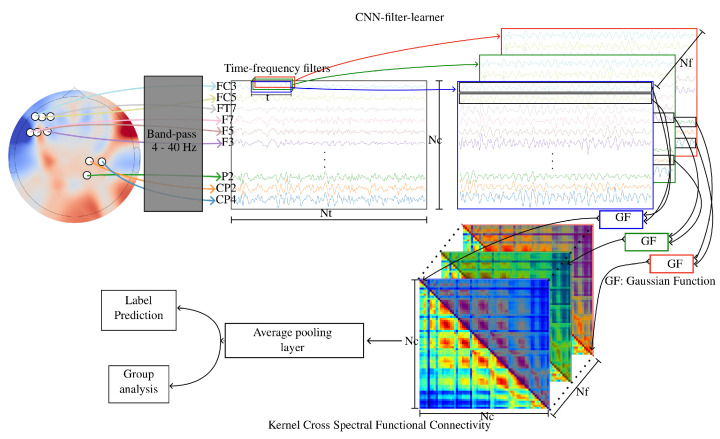
Kernel Cross-Spectral Functional Connectivity Network-(KCS-FCnet) for Motor Imagery Classification main sketch.

**Figure 4 diagnostics-13-01122-f004:**
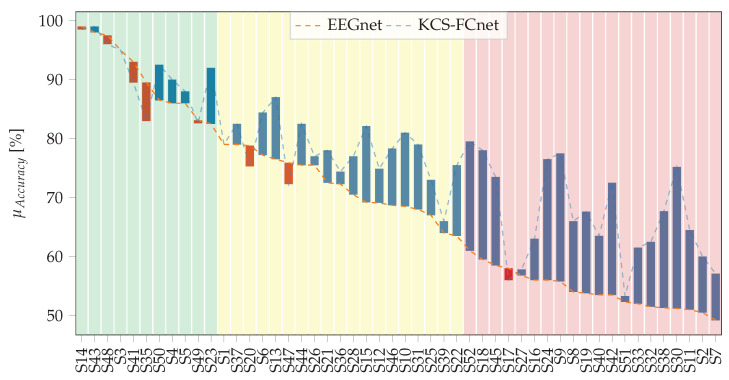
Subject specific results. EEGnet and KCS-FCnet average accuracies are depicted, with subjects being sorted based on their performance using EEGnet. The blue bars represent an improvement in performance using the KCS-FCnet, while the red bars indicate a decrease in performance. The background codes the group membership (best—G I, medium—GII, and worst—GIII performance clusters).

**Figure 5 diagnostics-13-01122-f005:**
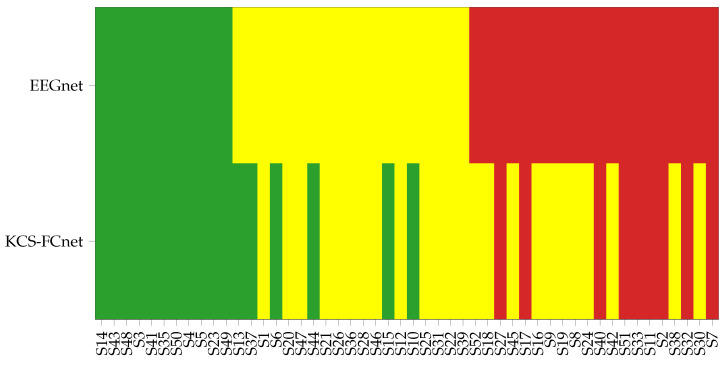
KCS-FCnet subject group enhancement regarding the EEGnet performance. Note that green, yellow and red represent best, medium, and worst performance regarding the average accuracy along subjects. First row: Subjects organized based on the EEGnet classification. Second row: Subject membership changes based on the KCS-FCnet results.

**Figure 6 diagnostics-13-01122-f006:**
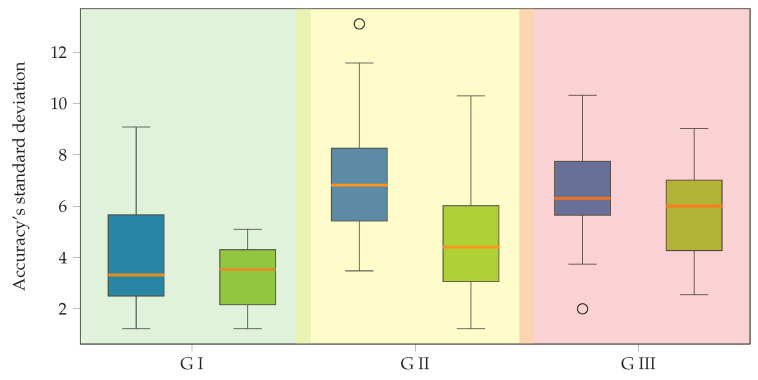
Group comparison between EEGnet (blue boxplots) and KCS-FCnet (green boxplots) concerning the accuracy’s standard deviation. The background codes the group membership (best—GI, medium—GII, and worst—GIII performance clusters).

**Figure 7 diagnostics-13-01122-f007:**
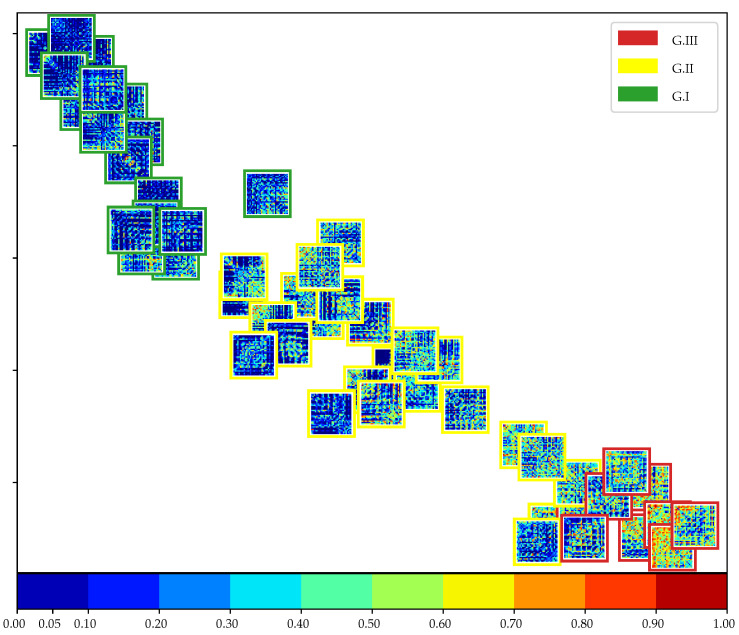
*t*-SNE 2D projection of pruned functional connectivity matrices based on KSC-FCnet and two-sample Kolmogorov–Smirnov test. The color bar depicts the *p*-value of every connection for each subject matrix, where deep blue means more class separability. Therefore, the bluer the matrix, the better the discriminability. Outer boxes indicate subject group belongingness: green G I, yellow G II, and red G III. *p*-values below 5% are taken as significant.

**Figure 8 diagnostics-13-01122-f008:**
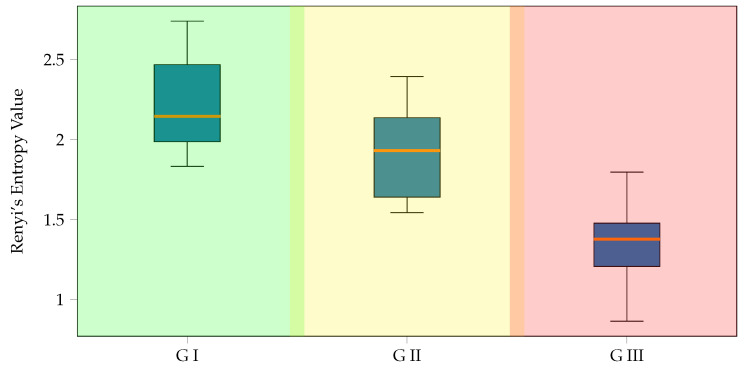
Rényi’s entropy-based retained information within the estimated functional connectivity matrices using KSC-FCnet. The background codes the group membership (best, medium, and worst performance clusters). Boxplot representation is used to present the retained information within each group.

**Figure 9 diagnostics-13-01122-f009:**
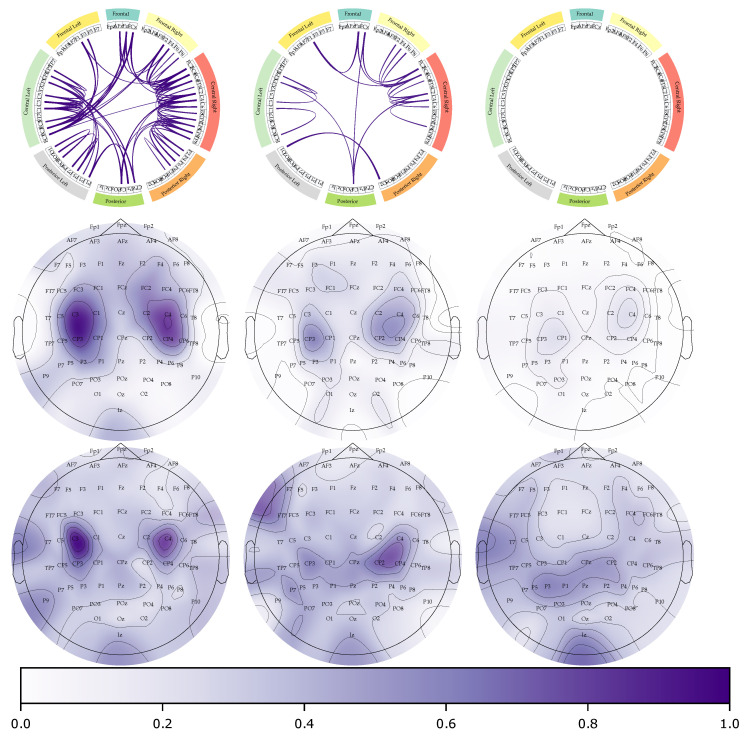
KSC-FCnet functional connectivity results (Circos plots and topoplots). The first row illustrates the 99th percentile of the most significant functional connections across the centroid subjects concerning the studied best, medium, and worst performance clusters (see Figure 2), with the opacity representing the strength of the connectivity. The second and third rows display topoplots of the two-sampled Kolmogorov–Smirnov test and the weights of the classification layer on the KCS-FCnet, respectively. The purer the purple color, the more important the brain area is. The left to right column represents each group from G I to G III.

**Figure 10 diagnostics-13-01122-f010:**
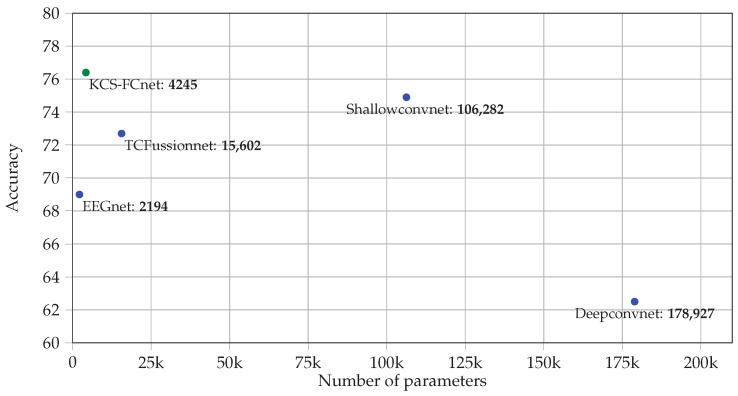
Method comparison results: number of trainable parameters vs. average motor imagery classification accuracy for the Giga database.

**Table 1 diagnostics-13-01122-t001:** Detailed KCS-FCnet architecture for MI classification.

Layer	Output Dimension	Params.
Input	$N_{c}\times N_{t}\times 1$	·
Conv2D	$N_{c}\times \left(N_{t}-\Delta_{t}+1\right)\times N_{f}$	max norm = 2.0, kernel size = (1, $\Delta_{t}$) Stride size = (1, 1), Bias = False
BatchNormalization	$N_{c}\times \left(N_{t}-\Delta_{t}+1\right)\times N_{f}$	·
ELU activation		
FCblock	$N_{f}\times \left(N_{c}\cdot \left(N_{c}-1\right)/2\right)\times 1$	·
AveragePooling2D	$1\times (N_{c}\cdot \left(N_{c}-1\right)/2)\times 1$	·
BatchNormalization	$1\times (N_{c}\cdot \left(N_{c}-1\right)/2)\times 1$	·
ELU activation		
Flatten	$N_{c}\cdot \left(N_{c}-1\right)/2$	·
Dropout	$N_{c}\cdot \left(N_{c}-1\right)/2$	Dropout rate = 0.5
Dense	$N_{y}$	max norm = 0.5
Softmax		

**Table 2 diagnostics-13-01122-t002:** Group-based accuracy results for EEGnet and KCS-FCnet. The average accuracy for the best, medium, and worst-grouped subjects is depicted. The KCS-FCnet average increase for each cluster is also reported.

Approach	Group	Accuracy	KCS-FCnet Gain
EEGnet	G I	90.6±4.3	·
	G II	72.2±7.3	·
	G III	54.3±6.6	·
KCS-FCnet	G I	** 91.5±3.3 **	**0.9**
	G II	** 77.8±4.7 **	**5.6**
	G III	** 66.7±5.6 **	**12.4**

**Table 3 diagnostics-13-01122-t003:** Method Comparison results regarding the average MI classification for the Giga database.

Approach	Accuracy	Kappa	AUC
Deepconvnet [21]	62.5±13.0	24.5±25.9	68.9±17.8
EEGnet [22]	69.0±14.6	38.0±29.1	75.4±16.6
TCFussionnet [23]	72.7±14.0	45.0±28.2	79.6±15.9
Shallowconvnet [21]	74.9±13.9	49.5±27.8	79.9±15.1
KCS-FCnet	** 76.4±11.3 **	** 52.6±22.7 **	** 82.2±12.2 **

## Data Availability

Publicly available datasets were analyzed in this study. This data can be found here: http://gigadb.org/dataset/100295 (accessed on 27 January 2023).

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
