# Peer review of "KCS-FCnet: Kernel Cross-Spectral Functional Connectivity Network for EEG-Based Motor Imagery Classification"

_diagnostics, 2023, doi:10.3390/diagnostics13061122_

Round 1

Reviewer 1 Report

1. Lines 108 - 111 mentioned detail of the motor imagery tasks. It would be better if an example of EEG signals and the associated motor imagery tasks was presented in the paper.

This will help readers to understand clearly the objective of the study.

2. There are a number of published research articles about the EEG signals of motor imagery classification. The reviewer suggests adding a comparison between the proposed method with some machine learning methods such as ANN or SVM.

Please find more literature and provide an objective comparison study.

For examples:

- The Implementation of EEG Transfer Learning Method Using Integrated Selection for Motor Imagery Signal

- Time Domain Features for EEG Signal Classification of Four Class Motor Imagery Using Artificial Neural Network

Reviewer 2 Report

Please find attached my comments to authors.

Round 2

Reviewer 1 Report

No further comments are available except the question marks that exist on the paper to indicate the citation. Please check carefully before publication.

Reviewer 2 Report

The paper can be published in its present form.